# The efficiency and safety of alendronate versus teriparatide for treatment glucocorticoid-induced osteoporosis: A meta-analysis and systematic review of randomized controlled trials

**Zhi-Ming Liu[1], Min Zhang[2], Yuan Zong[1], Ding Zhang[1], Zhu-Bin Shen[1], Xiao-Qing Guan[1], Fei Yin[1]\***

1 Department of Spine Surgery, China-Japan Union Hospital of Jilin University, Changchun, 130033, China,
2 Department of Neonatology, The Second Affiliated Hospital and Yuying Children's Hospital of Wenzhou Medical University, Wenzhou, 325027, China

\* yinfei999@jlu.edu.cn

## Abstract

### Background

Glucocorticoid-induced osteoporosis (GIOP) is the most common secondary osteoporosis, alendronate (ALE) and teriparatide (TPTD) are widely used in the treatment of GIOP. However, which of these two drugs has a better curative effect needs the support of evidence-based medicine.

### Methods

We searched PubMed, Embase, Cochrane Library, Web of Science, and Google Scholar for randomized controlled trials of ALE and TPTD in the treatment of glucocorticoid-induced osteoporosis until February 2022. These patients included in the study took glucocorticoid doses greater than 7.5 mg/d for more than 3 months before treatment with ALE and TPTD. The risk ratio (RR) and its 95% confidence interval (CI) are used as the influence index of discontinuous data, and the standardized mean difference (SMD) and its 95% CI are used as the influence index of continuous data.

### Results

A total of 4102 patients were enrolled in all 5 studies that met the admission criteria. We found that compared with ALE, TPTD could reduce the rate of new vertebral fracture (RR = 0.13, 95% CI: 0.05–0.34, P<0.00001). TPTD increased LS bone mineral density (BMD) (0.53, 95% CI 0.42–0.64, P<0.00001), TH BMD (0.17, 95% CI 0.05–0.28, P = 0.004) and FN BMD (0.17, 95% CI 0.05–0.29, P = 0.006) compared to ALE. However, there was no significant difference in the incidence of non-vertebral fracture and adverse events between the two groups.

**Data Availability Statement:** All relevant data are within the paper and its Supporting Information files.

**Funding:** The authors received no specific funding for this work.

**Competing interests:** The authors have declared that no competing interests exist.

## Conclusions

Compared with ALE, TPTD is an effective drug to reduce vertebral fracture risk in patients with GIOP. Furthermore, long-term use of TPTD can increase the bone mineral density of LS, FN, and TH.

## Introduction

Glucocorticoids are widely used in the treatment of rheumatic and autoimmune-related diseases (rheumatoid arthritis, systemic lupus erythematosus, polymyositis, vasculitis), inflammatory bowel disease, nephrotic syndrome, interstitial pneumonia, severe infection, shock, and so on. However, glucocorticoids cause severe side effects on bones, which makes GIOP the most common secondary osteoporosis [1]. In patients who receive long-term glucocorticoid treatment, 30–50% may have fractures, especially the lumbar spine (LS) femoral neck (FN), and total hip (TH) [2]. Taking 2.5 mg of oral prednisone daily increases the risk of fractures, and when the dose is greater than 7.5 mg (dosage equivalent to daily endogenous glucocorticoid production), the risk increases 5 times [3]. In addition, the risk of fractures increases significantly with the increasing dose of glucocorticoids and the prolongation of time [4].

The mechanism of GIOP can be summarized as follows: glucocorticoids reduce the number of osteoblasts and inhibit their function, stimulate the production of osteoclasts and increase their activity, thereby hindering bone growth and development. Glucocorticoid mainly interferes with bone formation by up-regulating peroxisome proliferator-activated receptor γ receptor 2 (PPARγ2) and affecting the Wnt/β-catenin signaling pathway. The former is beneficial to the differentiation of pluripotent precursor cells into adipocytes, resulting in a reduction in the number of osteoblasts. The latter is due to increased expression of sclerostin, inhibits Wnt signaling, resulting in reduced differentiation of osteoclast precursors into mature osteoblasts and increases apoptosis of osteoblasts and osteocytes [5].

The pathogenesis of GIOP is multifactorial, with both glucocorticoids' direct effects on osteocytes and indirect effects on multiple neuroendocrine and metabolic pathways.

The latter is mainly manifested by hypogonadism, decreased physical activity, increased calcium loss from the kidneys and intestines, and decreased production of growth hormone, insulin-like growth factor 1 (IGF1), and IGF1 binding protein (IGF-BP) [6]. In addition, excessive use of glucocorticoids can adversely affect bones and muscles, causing bone and muscle atrophy and weakness, and increasing the risk of falls and fractures [7].

Bisphosphonate is a synthetic pyrophosphate analog, which can inhibit osteoclast activity, inhibit bone resorption and increase bone mineral density in patients treated with glucocorticoid. It has been proved to be an effective method for the prevention and treatment of GIOP. ALE is bisphosphate that can effectively increase the BMD of the LS, FN, and TH. It has been widely used to prevent and treat GIOP [8]. TPTD, as a parathyroid hormone analog, can effectively induce pre-osteoblasts to differentiate into osteoblasts, improve osteoblasts' activity, stimulate pre-existing osteoblasts to form new bone, and reduce osteoblasts apoptosis [9].

The meta-analysis previously published did not focus on the specific disease of glucocorticoid-induced osteoporosis, nor two specific drugs, nor did it discuss the spines that exert drug effects at the micro-level such as bone metabolism indicators, nor did it pay more attention to the main adverse reactions of drugs. The main purpose of this systematic review and meta-analysis is to compare the safety and effectiveness of TPTD and ALE and to provide a new idea for the clinical treatment of GIOP.

## Methods

This study is reported by PRISMA (Preferred Reporting Items for Systematic Reviews and Meta-Analyses). Please refer to S1 file for detailed table contents.

### Search strategy and selection criteria

We searched PubMed, Embase, Cochrane Library, Web of Science, and Google databases for randomized controlled trials of ALE and TPTD in the treatment of glucocorticoid-induced osteoporosis until February 2022. In the retrieval process, we use keywords and Medical Subject Headings (MESH) terms to search the database. Search strategy is as follows: ("alendronate"[MeSH Terms] AND "teriparatide"[MeSH Terms]) AND ((("glucocorticoid "[MeSH Terms]) AND "osteoporosis"[MeSH Terms) OR "glucocorticoid-induced osteoporosis "[All Fields] OR "GIOP "[All Fields]])). In this meta-analysis, all data are extracted from previously published studies, so patient consent and ethical approval are not required. Specific literature search strategies can refer to S2 File.

### Study inclusion and exclusion criteria

The inclusion criteria are as follows: (1)Patients were at least 21 years old; (2) Patients had taken prednisone or its equivalent at a dosage of $\geq 5$ mg/day for$\geq 3$ months before screening; (3) Patients were required to have an LS or TH BMD T score of $\leq -2.0$ or $\leq -1.0$ plus at least one fragility fracture while taking glucocorticoids; (4) Studies' language was English; (5) Studies were RCTs.

The exclusion criteria are as follows:(1) Primary osteoporosis (including postmenopausal osteoporosis, senile osteoporosis, and idiopathic osteoporosis) and other secondary osteoporosis caused by non-glucocorticoid; (2) The type of articles was review, meta-analysis, and other non-RCT; (3) The content and outcome are not the incidences of vertebral fracture and the change of BMD.

### Data extraction and quality assessment

ZML, YZ, DZ, XQG, and ZBS were responsible for the literature No. 11–14, including extraction of relevant content in this article, including major and minor outcomes. ZML was also responsible for collecting and integrating the data extracted by the five authors.

The primary outcome of this study was the incidence of vertebral fracture and non-vertebral fracture, while the secondary outcome was the mean percent changes from baseline to 6,12,18 months in the ALE and TPTD groups in the lumbar spine(LS) bone mineral density (BMD), the mean percent changes from baseline to 18 months in the two groups in the BMD of the femoral neck (FN) and total hip (TH), the incidence of 5 adverse events with the highest incidence, and changes in bone formation and resorption markers in both groups. In this paper, two researchers independently conducted the literature search, screening, data extraction, and heterogeneity analysis. If there is any objection, we will reach an agreement after discussion, complete the preliminary search according to the established search strategy, and read the abstract and full text to exclude studies that do not meet the inclusion criteria.

### Data synthesis and analysis

All data are summarized using mean and standard deviation (SD). We use risk ratio (RR) with a 95% confidence interval (CI) as the effective index of discontinuous data and standardized mean difference (SMD) as the effective index of continuous data. We use $I^2$ to evaluate heterogeneity between studies, and if $I^2 \geq 50\%$ is considered to be high heterogeneity, a random

effect model is used, otherwise, a fixed-effect model is used. All of the above analysis was done through Review Manager version 5.3. Egger linear regression test and funnel plot were performed using Stata 16.0 to estimate publication bias. Specific data processing processes and code can refer to S2 File.

## Results

### Search results and study characteristics

Of the 62 articles initially searched, 30were excluded because of content and outcome irrelevant, 15were excluded because of repetition, and17 were excluded because of non-RCTs. A total of 4102patients were included in the 5 studies [10–14] that met the inclusion criteria of this article (**Fig 1**).

We counted the baseline characteristics of these studies, including the studies' first author and year of publication, the number of patients in each study, the dose and duration of glucocorticoids taken, and gender, age, menopause (female), and BMD or T-score of the LS and FN before taking the drug (**Table 1**).

### Risk of bias in the included studies

All studies are randomized controlled trials but do not explain the random sequence generation process, and some of them have reporting biases (**Fig 2**).

The Egger linear regression test and funnel plot were used to measure the percent change in LS BMD at 18 months. The funnel plot does have certain asymmetry, but each study is within a 95% confidence interval, and the Egger linear regression test P = 0.07 > 0.05, which shows that publication bias is not statistically significant, but the number of studies included in this study is limited, and the interpretation of the results should be more cautious (**Fig 3**).

### Incidence of vertebral and non-vertebral fractures

We found that compared with ALE, TPTD could significantly reduce the rate of new vertebral fracture (RR = 0.13, 95% CI: 0.05–0.34, P<0.00001), but there was no significant difference between the two in terms of non-vertebral fractures (RR = 1.28, 95% CI: 0.81–2.02, P = 0.29) (**Figs 4 and 5**).

### Mean percent change from baseline in the BMD at the LS, FN, TH

We first analyzed the percentage change of LS BMD at 6, 12, and 18 months. The results showed that compared with ALE, TPTD could increase LS BMD from baseline to 6 months (0.30, 95% CI 0.19–0.42, P<0.00001), 12 months (0.48, 95% CI 0.36–0.60, P<0.00001) and 18 months (0.53, 95% CI 0.42–0.64, P<0.00001) (**Fig 6**).

Compared with ALE, TPTD increased the BMD of FN from baseline to 18 months (0.17, 95% CI 0.05–0.29, P = 0.006); and increased the BMD of TH from baseline to 18 months (0.17, 95% CI 0.05–0.28, P = 0.004) (**Figs 7 and 8**).

We further analyzed that the longer the time of taking TPTD within a certain period, the more obvious the increase of LS BMD, and the LS BMD at 18 months was significantly higher than that at 6 months and 12 months. In addition, TPTD had different effects on different parts bone tissue. After taking TPTD for 18 months, the increase of BMD of LS was higher than that of FN and TH.

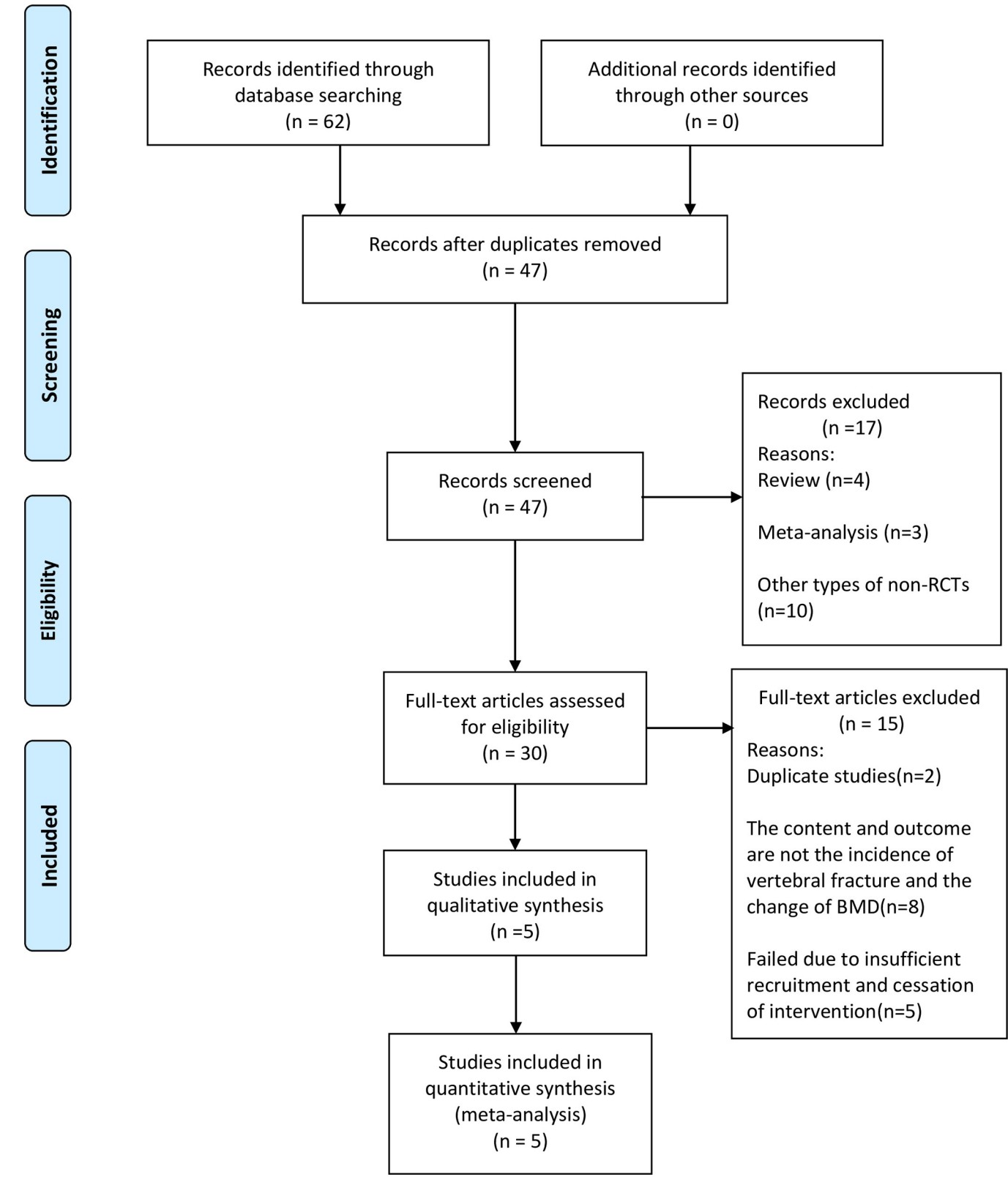

**Fig 1. Flow diagram of literature search and study inclusion.**

**Table 1. Baseline characteristics[a].**

| Comparison | N | GC dose(mg/d)[b] | GC duration (m) | Age(y) | Sex (M/F) | Postmenopause n(%) | LS BMD(gm/cm2)/ T-score | PINP (ug/l) | CTX(pmol/l) |
|---|---|---|---|---|---|---|---|---|---|
| **Benito R. Losada 2008[d]** | | | | | | | | | |
| alendronate | 32 | 7.5±1.7 | 5.3±2.9 | 54.9±4.5 | 5/27 | NM | 0.8 ±0.05 | NM | NM |
| teriparatide | 29 | 8.8±1.9 | 2.7±3.2 | 52.5 ±5.0 | 5/24 | NM | 0.8 ±0.05 | NM | NM |
| **Alan L. Burshell 2009[d]** | | | | | | | | | |
| alendronate | 77 | 8.0 | 16.8 | 60.6±2.5 | 17/60 | 50(64.9) | −2.7±0.1 | 39.7±4.5 | 3570.8 ±616.6 |
| teriparatide | 80 | 7.5 | 14.4 | 56.1±2.6 | 13/67 | 41(51.3) | −2.5±0.1 | 44.5±4.8 | 3585.0 ±643.2 |
| **Jean-Pierre 2009[d]** | | | | | | | | | |
| alendronate | 192 | 10.1±0.7 | 5.1 ± 0.5 | 57.1±1.0 | NM | NM | 0.85±0.01 | NM | NM |
| teriparatide | 195 | 9.4±0.4 | 5.2 ± 0.6 | 55.8±1.0 | NM | NM | 0.85±0.01 | NM | NM |
| **B. L. Langdahl 2009[d]** | | | | | | | | | |
| alendronate | | | | | | | | | |
| Postmenopausal | 143 | 7.3 | 26.4 | 62.1±1.2 | 0/143 | 143(100) | −2.7±0.1 | 39.0±4.7 | 3844.8 ±580.5 |
| Premenopausal | 30 | 10.0 | 10.8 | 35.8±2.1 | 0/30 | 0 | −2.6±0.2 | 43.0±3.6 | 2670.0 ±217.6 |
| Men | 41 | 10.0 | 25.2 | 59.7±1.9 | 41/0 | 0 | −2.3±0.2 | 36.5±9.6 | 4173.8 ±977.4 |
| teriparatide | | | | | | | | | |
| Postmenopausal | 134 | 7 | 31.2 | 61.9±1.2 | 0/134 | 134(100) | −2.7±0.1 | 47.3±6.4 | 4030.8 ±606.5 |
| Premenopausal | 37 | 8 | 21.6 | 40.0±1.9 | 0/37 | 0 | −2.4±0.2 | 34.8±4.2 | 2331.0 ±602.0 |
| Men | 42 | 10 | 27.6 | 55.5±1.9 | 42/0 | 0 | −2.3±0.2 | 36.3±8.0 | 3236.0 ±837.8 |
| **Kenneth G. Saag 2009[d]** | | | | | | | | | |
| alendronate | 214 | ≥5 | 24 | 57.3 ±14.0 | 41/173 | 143 (66.8) | 0.864±0.014 | 39.5±4.0 | 3604.0 ±540.1 |
| teriparatide | 214 | ≥5 | 27.6 | 56.1 ±13.4 | 42/172 | 134 (62.6) | 0.863±0.014 | 41.0±5.1 | 3384.5 ±486.4 |

[a]All patients received supplements of calcium (1000 mg/d) and vitamin D (800 IU/d).

[b] prednisone or equivalent.

[c]values are mean±SD.

[d]values are mean±SE.

BMD = bone mineral density.

LS = lumbar spine.

PINP = N-terminal propeptide of type I collagen.

CTX = C-telopeptide of type I collagen.

GC = glucocorticoid.

M = male.

F = female.

NM = not mentioned.

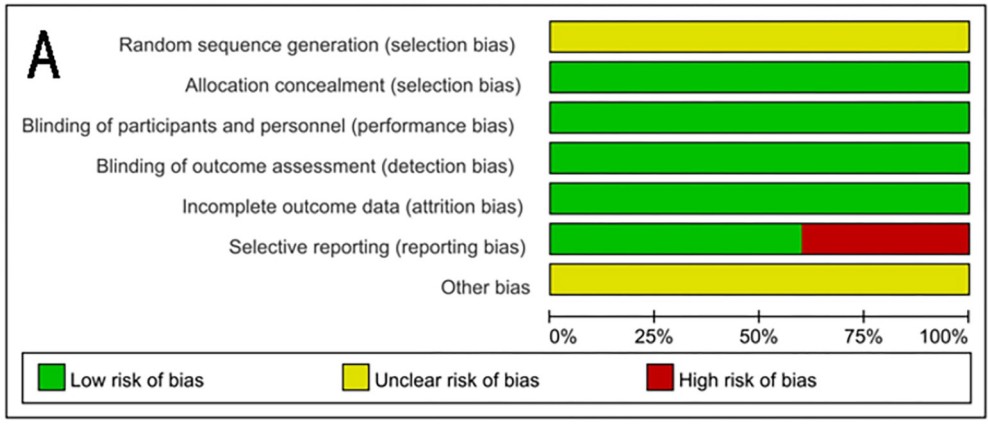

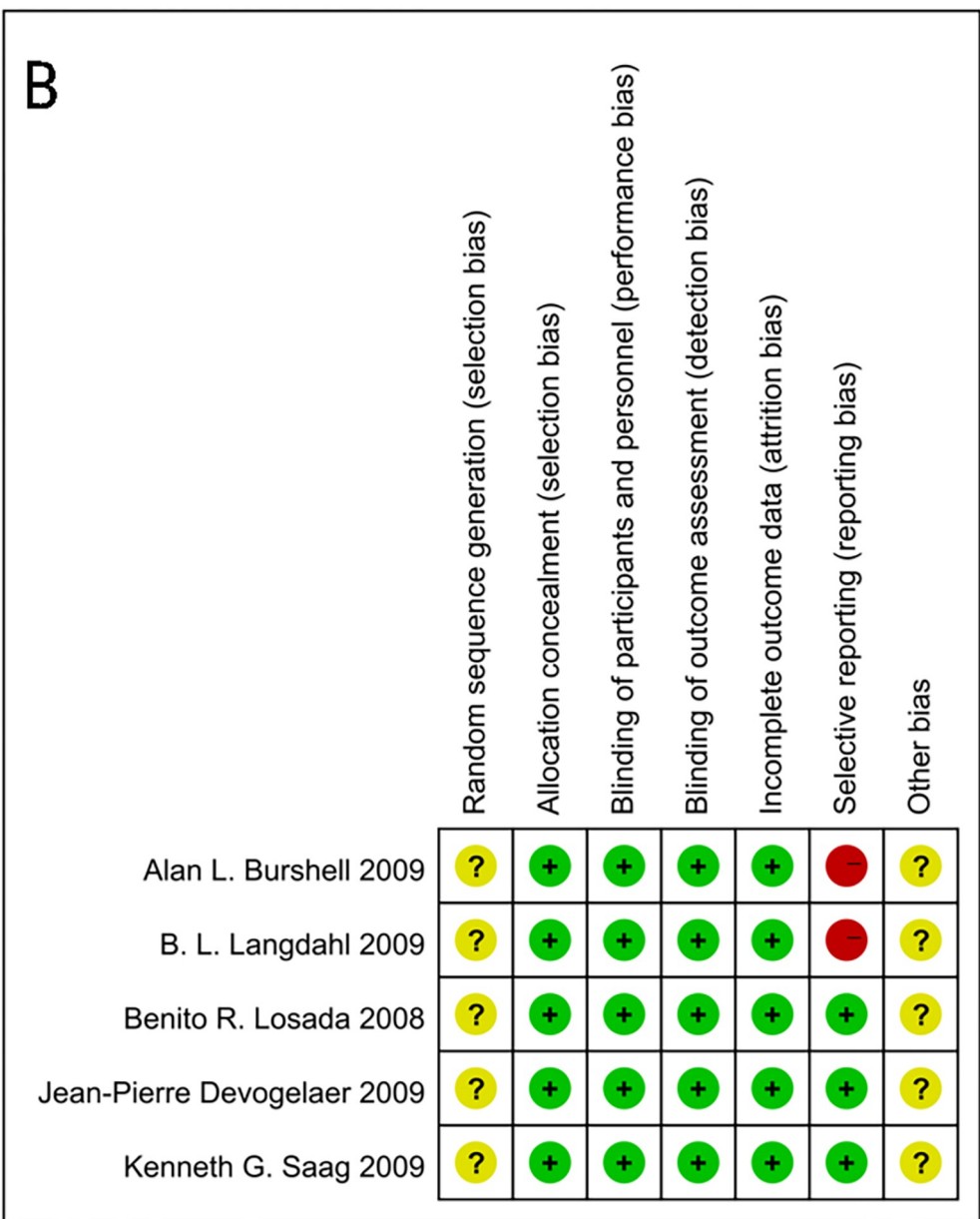

**Fig 2. Risk of bias assessment of each included study.** (A) Risk of bias graph. (B) Risk of bias summary.

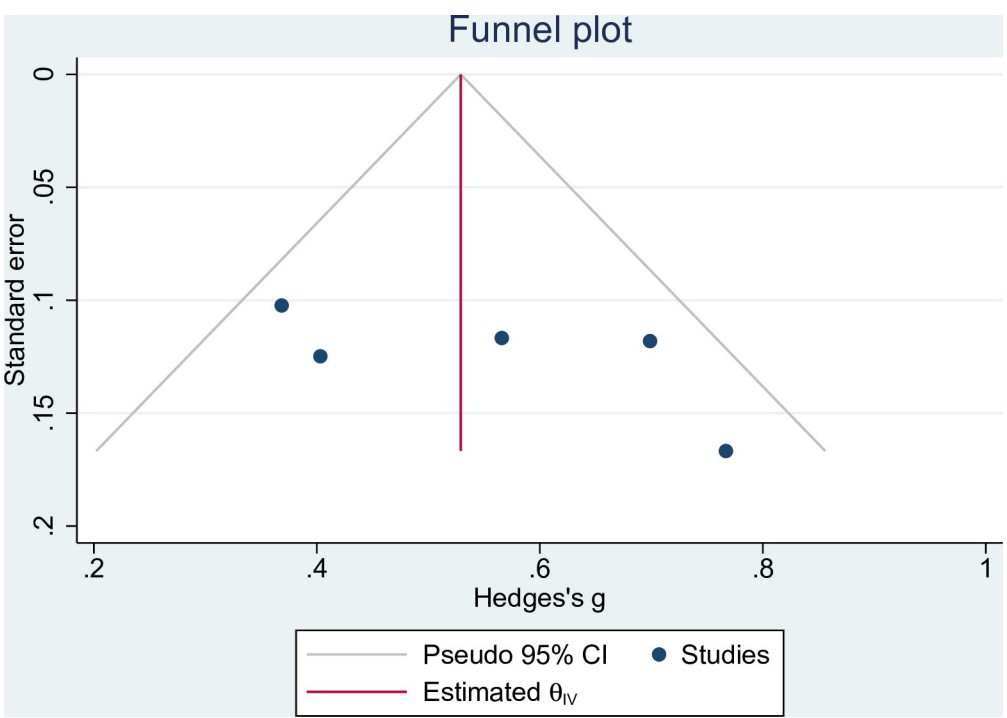

**Fig 3. Funnel plot of percent change in LS BMD at 18 months with alendronate and teriparatide.**

## Percent change in markers of bone formation and resorption

Two studies have reported bone metabolic markers, of which N-terminal propeptide of type I collagen (PINP) is a marker of bone formation and C-telopeptide of type I collagen (CTX) is a marker of bone resorption. In patients taking TPTD, both PINP and CTX increased compared to the baseline, began to rise at 1 month, peaked at 6 months, and decreased slowly at 18 months. However, in patients taking ALE, both PINP and CTX decreased from the baseline, began to decline at 1 month, reached the lowest level at 6 months, and increased slowly at 18 months (**Figs 9 and 10**).

## Adverse events

A total of 3 studies reported the incidence of adverse reactions to the two drugs, we selected 5 high-incidence adverse events for research. The results showed that, overall, there was no significant difference in the incidence of adverse reactions between the two groups (RR = 1.00,

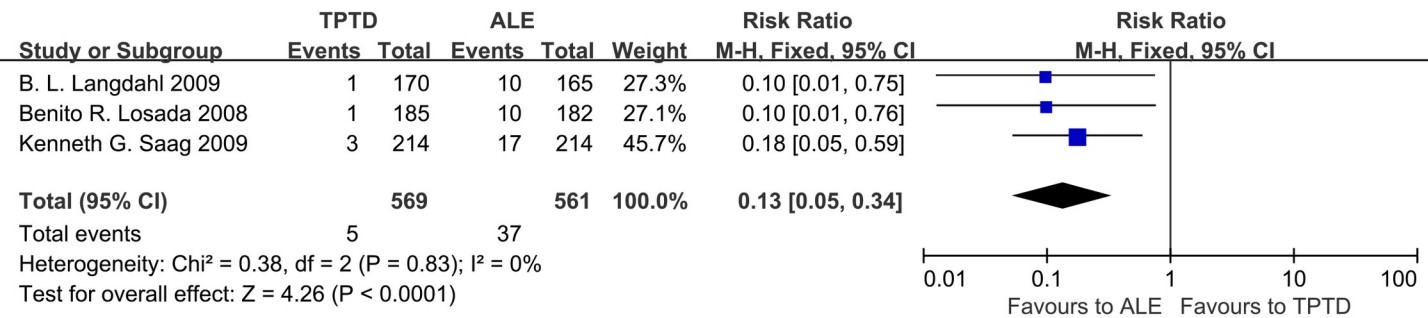

**Fig 4. Forest plots of incidence of vertebral fractures.**

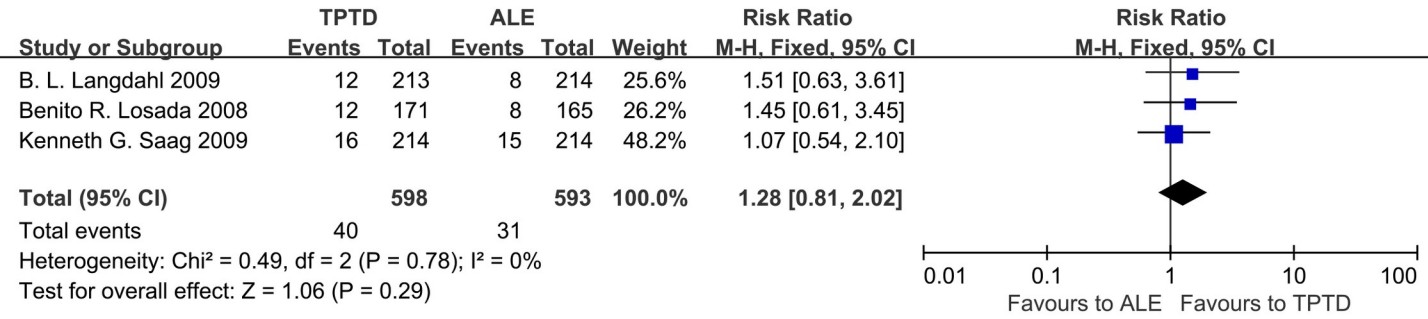

Fig 5. Forest plots of incidence non-vertebral fractures.

95% CI: 0.89–1.12, P = 0.58). Patients taking TPTD had a higher incidence of nausea than ALE (RR = 1.68, 95% CI: 1.19–2.36, P = 0.003). In contrast, patients taking ALE had a higher incidence of dyspepsia (RR = 0.51, 95% CI: 0.31–0.83, P = 0.007) and urinary tract infections (RR = 0.69, 95% CI: 0.48–0.99, P = 0.04) than TPTD (**Fig 11**).

## Discussion

Through meta-analysis and systematic review of 5 randomized controlled trials, we found that compared with ALE, TPTD can effectively increase the BMD of LS, FN, and TH, and the

Fig 6. Forest plots of mean percent change from baseline to 6, 12, and 18 months in the LS BMD.

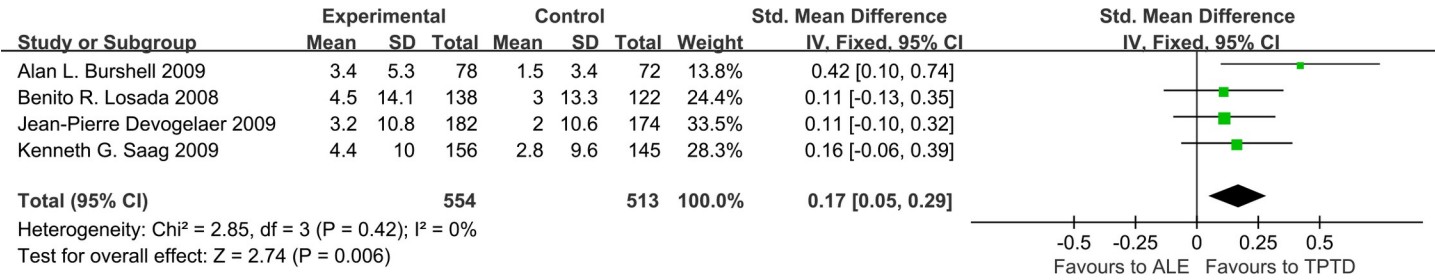

**Fig 7. Forest plots of mean percent change from baseline to 18 months in the FN BMD.**

incidence of vertebral fracture was lower. However, there was no significant difference in the incidence of non-vertebral fracture and adverse reactions between the two groups.

Consistent with the results of this article, Ya-Kang Wang et al. [15] conducted a meta-analysis of ALE for GIOP and found that ALE can significantly increase the BMD of the LS and FN. Similarly, Chun-Lin Liu et al. [9] conducted a meta-analysis of bisphosphonates and TPTD for osteoporosis and found that TPTD can significantly increase the BMD of LS, TH, and FN, especially GIOP; and compared with bisphosphonates, TPTD cannot reduce the incidence of non-vertebral fractures.

Another meta-analysis showed that ALE, as a second-generation bisphosphonate, could significantly increase the BMD of LS, TH, and FN, and the incidence of gastrointestinal adverse reactions was very low, but could not reduce the incidence of vertebral and non-vertebral fractures [16]. But our study shows that TPTD is more effective than ALE in reducing the incidence of vertebral fractures, which reflects the unique advantages of TPTD.

Because of the high price and gastrointestinal adverse reactions, TPTD is used as a second-line drug, but the results show that compared with ALE, it can effectively reduce the incidence of vertebral fracture, and there is no significant difference in adverse reactions. In addition, TPTD also has its unique advantages, as a synthetic metabolic agent, it is significantly better than bisphosphonate in preventing glucocorticoid-induced bone loss and fracture, and can reduce the adverse reactions of glucocorticoids, which is consistent with the results of this study. However, the guidelines do not yet use TPTD as a first-line drug [2].

Through meta-analysis and systematic evaluation of five randomized controlled trials, the main outcomes were as follows: Compared with ALE, patients taking TPTD had a lower incidence of vertebral fracture; secondary outcomes: similarly, patients taking TPTD improved the BMD of LS, FN, and TH compared to those taking ALE, and the BMD of LS, FN, and TH gradually increased with the prolongation of dosing time. There was no statistical difference between the two drugs in the incidence of non-vertebral fractures and adverse reactions.

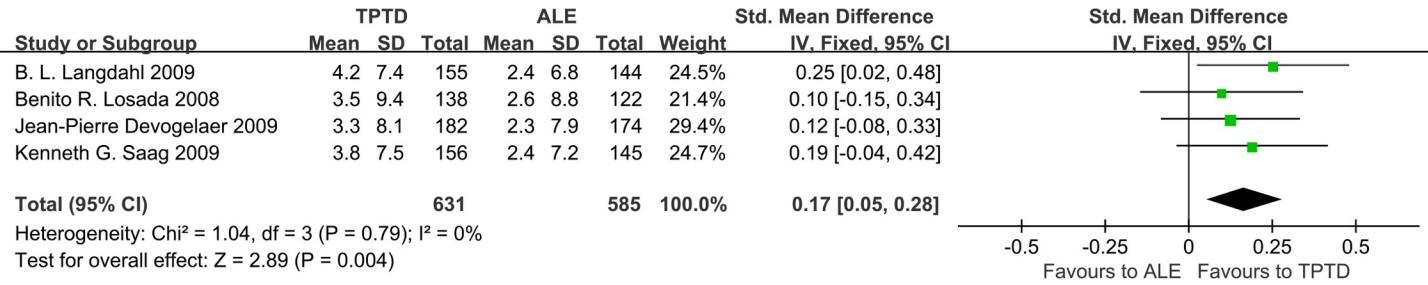

**Fig 8. Forest plots of mean percent change from baseline to 18 months in the TH BMD.**

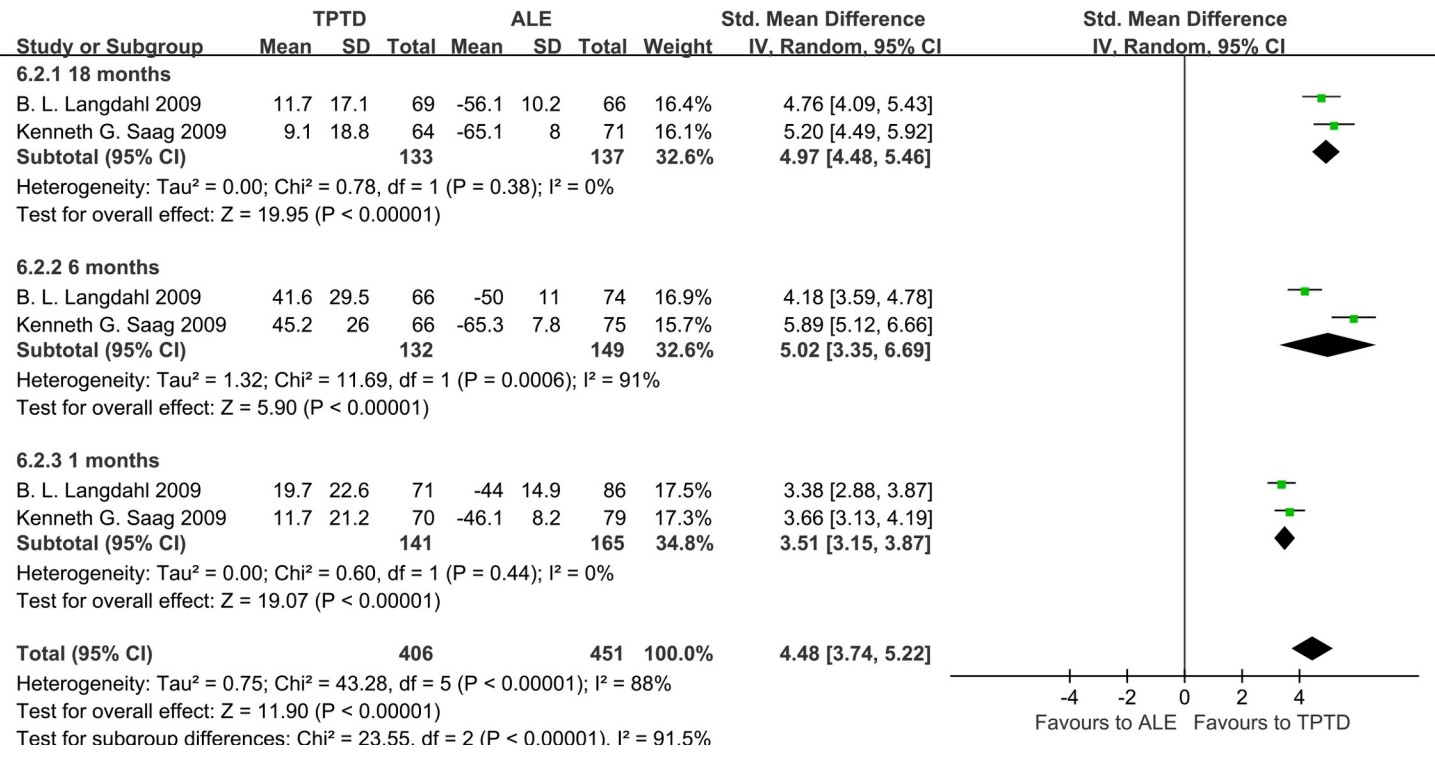

Fig 9. Forest plots of percentage changes of bone formation marker PINP.

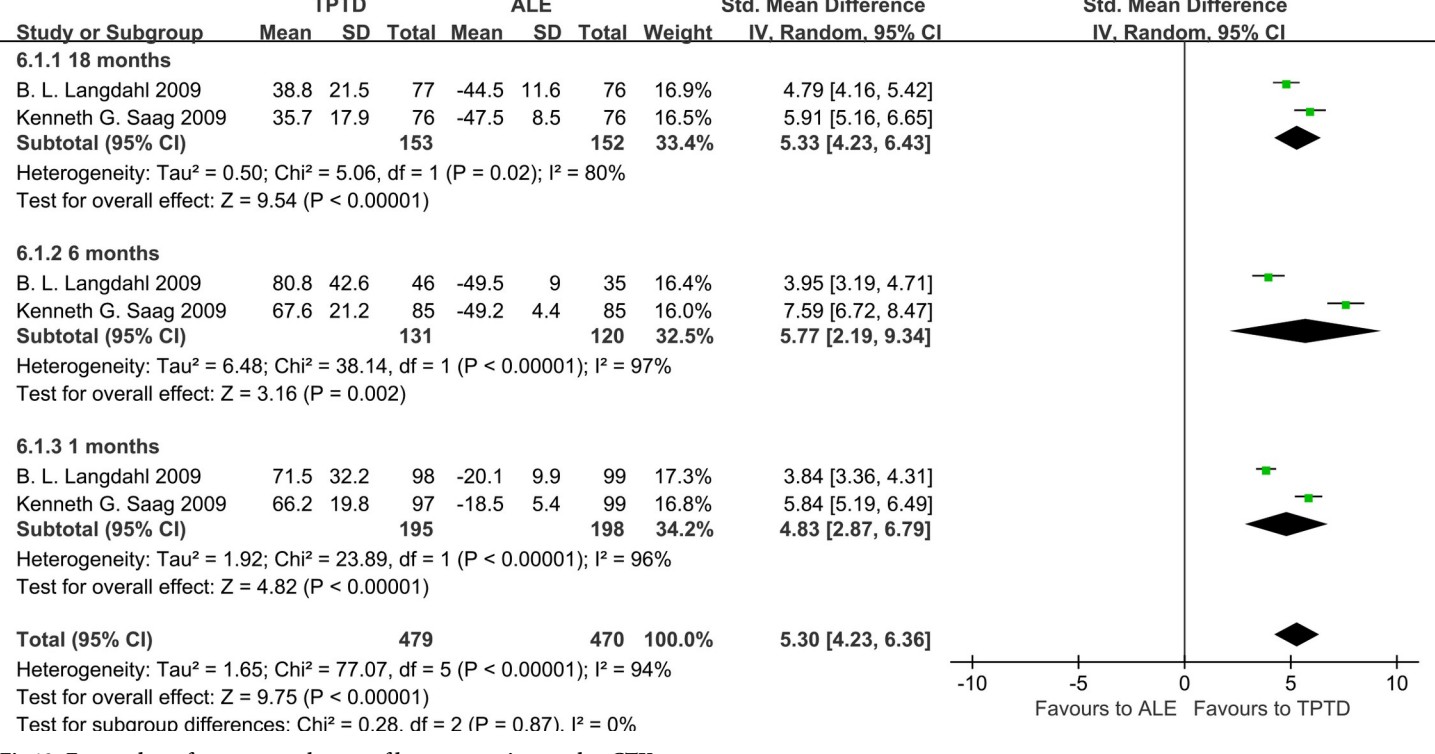

Fig 10. Forest plots of percentage changes of bone resorption marker CTX.

| Study or Subgroup | TPTD Events | Total | ALE Events | Total | Weight | Risk Ratio M-H, Fixed, 95% CI | Risk Ratio M-H, Fixed, 95% CI |
|---|---|---|---|---|---|---|---|
| **5.1.1 Incidence of all main adverse events** | | | | | | | |
| B. L. Langdahl 2009 | 83 | 213 | 84 | 214 | 18.0% | 0.99 [0.78, 1.26] | |
| Benito R. Losada 2008 | 69 | 185 | 72 | 182 | 15.6% | 0.94 [0.73, 1.22] | |
| Kenneth G. Saag 2009 | 86 | 214 | 76 | 214 | 16.4% | 1.13 [0.89, 1.44] | |
| Subtotal (95% CI) | | 612 | | 610 | 50.0% | 1.02 [0.89, 1.18] | |
| Total events | 238 | | 232 | | | | |
| Heterogeneity: Chi² = 1.10, df = 2 (P = 0.58); I² = 0% | | | | | | | |
| Test for overall effect: Z = 0.31 (P = 0.76) | | | | | | | |
| | | | | | | | |
| **5.1.2 Nausea** | | | | | | | |
| B. L. Langdahl 2009 | 29 | 213 | 15 | 214 | 3.2% | 1.94 [1.07, 3.52] | |
| Benito R. Losada 2008 | 24 | 185 | 14 | 182 | 3.0% | 1.69 [0.90, 3.16] | |
| Kenneth G. Saag 2009 | 26 | 214 | 18 | 214 | 3.9% | 1.44 [0.82, 2.55] | |
| Subtotal (95% CI) | | 612 | | 610 | 10.1% | 1.68 [1.19, 2.36] | |
| Total events | 79 | | 47 | | | | |
| Heterogeneity: Chi² = 0.50, df = 2 (P = 0.78); I² = 0% | | | | | | | |
| Test for overall effect: Z = 2.95 (P = 0.003) | | | | | | | |
| | | | | | | | |
| **5.1.3 Headache** | | | | | | | |
| B. L. Langdahl 2009 | 16 | 213 | 12 | 214 | 2.6% | 1.34 [0.65, 2.76] | |
| Benito R. Losada 2008 | 14 | 185 | 9 | 182 | 2.0% | 1.53 [0.68, 3.45] | |
| Kenneth G. Saag 2009 | 19 | 214 | 14 | 214 | 3.0% | 1.36 [0.70, 2.64] | |
| Subtotal (95% CI) | | 612 | | 610 | 7.5% | 1.40 [0.92, 2.12] | |
| Total events | 49 | | 35 | | | | |
| Heterogeneity: Chi² = 0.07, df = 2 (P = 0.97); I² = 0% | | | | | | | |
| Test for overall effect: Z = 1.56 (P = 0.12) | | | | | | | |
| | | | | | | | |
| **5.1.4 Back pain** | | | | | | | |
| B. L. Langdahl 2009 | 18 | 213 | 22 | 214 | 4.7% | 0.82 [0.45, 1.49] | |
| Benito R. Losada 2008 | 15 | 185 | 19 | 182 | 4.1% | 0.78 [0.41, 1.48] | |
| Subtotal (95% CI) | | 398 | | 396 | 8.8% | 0.80 [0.52, 1.24] | |
| Total events | 33 | | 41 | | | | |
| Heterogeneity: Chi² = 0.02, df = 1 (P = 0.90); I² = 0% | | | | | | | |
| Test for overall effect: Z = 1.00 (P = 0.32) | | | | | | | |
| | | | | | | | |
| **5.1.5 Dyspepsia** | | | | | | | |
| B. L. Langdahl 2009 | 7 | 213 | 15 | 214 | 3.2% | 0.47 [0.20, 1.13] | |
| Benito R. Losada 2008 | 7 | 185 | 15 | 182 | 3.3% | 0.46 [0.19, 1.10] | |
| Kenneth G. Saag 2009 | 9 | 214 | 15 | 214 | 3.2% | 0.60 [0.27, 1.34] | |
| Subtotal (95% CI) | | 612 | | 610 | 9.7% | 0.51 [0.31, 0.83] | |
| Total events | 23 | | 45 | | | | |
| Heterogeneity: Chi² = 0.25, df = 2 (P = 0.88); I² = 0% | | | | | | | |
| Test for overall effect: Z = 2.70 (P = 0.007) | | | | | | | |
| | | | | | | | |
| **5.1.6 Urinary tract infection** | | | | | | | |
| B. L. Langdahl 2009 | 13 | 213 | 20 | 214 | 4.3% | 0.65 [0.33, 1.28] | |
| Benito R. Losada 2008 | 9 | 185 | 15 | 182 | 3.3% | 0.59 [0.27, 1.31] | |
| Kenneth G. Saag 2009 | 22 | 214 | 29 | 214 | 6.2% | 0.76 [0.45, 1.28] | |
| Subtotal (95% CI) | | 612 | | 610 | 13.8% | 0.69 [0.48, 0.99] | |
| Total events | 44 | | 64 | | | | |
| Heterogeneity: Chi² = 0.30, df = 2 (P = 0.86); I² = 0% | | | | | | | |
| Test for overall effect: Z = 2.02 (P = 0.04) | | | | | | | |
| | | | | | | | |
| Total (95% CI) | | 3458 | | 3446 | 100.0% | 1.00 [0.89, 1.12] | |
| Total events | 466 | | 464 | | | | |
| Heterogeneity: Chi² = 25.52, df = 16 (P = 0.06); I² = 37% | | | | | | | |
| Test for overall effect: Z = 0.02 (P = 0.99) | | | | | | | |
| Test for subgroup differences: Chi² = 23.56, df = 5 (P = 0.0003), I² = 78.8% | | | | | | | |

0.2    0.5    1    2    5
Favours to ALE   Favours to TPTD

**Fig 11. Forest plots of five high incidence adverse events.**

Ya-Kang Wang et al. [15] conducted a meta-analysis on the efficacy of ALE in the treatment of GIOP and found that ALE could significantly improve the BMD of LS and FN. After 12 months of medication, the BMD of LS increased without significant gastrointestinal adverse reactions. The results of these studies are consistent with this study. In addition, they also found that the fracture risk of patients who took ALE for 12 months did not change significantly, but our study found that ALE could reduce the incidence of vertebral fractures.

Chun-Lin Liu et al. [9] compared the efficacy of bisphosphonates and TPTD in the treatment of osteoporosis and found that TPTD could significantly increase the BMD of LS, TH, and FN in osteoporosis patients, especially GIOP. In addition, there was no difference in the effect of TPTD on the incidence of non-vertebral fractures when compared to bisphosphonates. The results of this study are consistent with our study.

Shun-Li Kan et al. [16] performed a meta-analysis on the efficacy of ALE in preventing GIOP in rheumatic patients and found that ALE could increase the BMD of LS, TH, and trochanter, which were consistent with the results of our study. There were no significant differences in the incidence of gastrointestinal adverse reactions and vertebral and non-vertebral fractures, in contrast, our study found that both TPTD and ALE were found to reduce the incidence of vertebral fractures. Compared with the previously published meta-analysis, this study differs in that it focuses mainly on glucocorticoid-induced osteoporosis rather than on osteoporosis in general. In our study, two of the two anti-osteoporosis drugs were selected to compare the efficacy and safety of GIOP, so it was more targeted. The main outcome of this study was the incidence of vertebral fracture, which was also more targeted. In addition, this study also had unique features that previous studies did not have. In our study, bone metabolism indexes were innovatively taken as evaluation indexes of drug efficacy, so that the mechanism of drug action can be explored from a microscopic perspective. In this study, the main adverse reactions of drugs were also used as evaluation indicators, instead of focusing on gastrointestinal reactions as in the previous study.

## Limitations

However, there are still some shortcomings in our study. First, whether the female subjects are postmenopausal or not may have an impact on the efficacy of drug treatment, but the women included in this study include both menopausal and non-menopausal women. Second, all the patients included in this study were given long-term additional calcium and vitamin D supplementation, which also had a certain impact on the efficacy of drug treatment. Third, the longest study included in this article is only 36 months, and the long-term efficacy of the drug needs to be further explored.

## Conclusion

In general, compared with ALE, TPTD can reduce the incidence of vertebral fracture, increase the BMD of LS, FN, and TH, and increase the bone metabolic markers. However, there was no significant difference in the incidence of non-vertebral fracture and main adverse reactions between the two groups.

## Supporting information

**S1 File. PRISMA_2020_checklist.**
(PDF)

**S2 File. Specific literature search strategies, data processing procedures and codes.**
(ZIP)

## Acknowledgments

The author thanks Min Zhang for polishing the language of this article.

## Author Contributions

**Conceptualization:** Zhi-Ming Liu, Ding Zhang, Zhu-Bin Shen.

**Data curation:** Zhi-Ming Liu, Yuan Zong.

**Formal analysis:** Zhi-Ming Liu, Min Zhang.

**Investigation:** Zhi-Ming Liu, Xiao-Qing Guan.

**Methodology:** Zhi-Ming Liu, Xiao-Qing Guan.

**Project administration:** Zhi-Ming Liu.

**Resources:** Min Zhang, Zhu-Bin Shen.

**Software:** Yuan Zong, Ding Zhang, Zhu-Bin Shen.

**Supervision:** Fei Yin.

**Validation:** Fei Yin.

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
