## [Decision Letter · Decision Letter 0]

17 Feb 2022

PONE-D-22-00100The efficiency and safety of alendronate versus teriparatide for treatment glucocorticoid-induced osteoporosis: A meta-analysis and systematic review of randomized controlled trialsPLOS ONE

Dear Dr. Fei Yin,

Thank you for submitting your manuscript to PLOS ONE. After careful consideration, we feel that it has merit but does not fully meet PLOS ONE’s publication criteria as it currently stands. Therefore, we invite you to submit a revised version of the manuscript that addresses the points raised during the review process.

ACADEMIC EDITOR: The reviewers have raised a number of points which we believe major modifications are necessary to improve the manuscript, taking into account the reviewers' remarks. Please consider and address each of the comments raised by the reviewers before resubmitting the manuscript. This letter should not be construed as implying acceptance, as a revised version will be subject to re-review.

We look forward to receiving your revised manuscript.

Kind regards,

Wisit Cheungpasitporn, MD

Academic Editor

PLOS ONE

Journal Requirements:

"This research received no specific grant from any funding agency in the public, commercial or not-for-profit sectors."

"All other authors have no conflicts of interest."

Reviewers' comments:

Reviewer's Responses to Questions

**Comments to the Author**

1. Is the manuscript technically sound, and do the data support the conclusions?

Reviewer #1: Yes

Reviewer #2: Partly

2. Has the statistical analysis been performed appropriately and rigorously? 

Reviewer #1: Yes

Reviewer #2: I Don't Know

3. Have the authors made all data underlying the findings in their manuscript fully available?

Reviewer #1: No

Reviewer #2: Yes

4. Is the manuscript presented in an intelligible fashion and written in standard English?

Reviewer #1: No

Reviewer #2: No

5. Review Comments to the Author

Reviewer #1: The authors present a comprehensive summary of their systematic review and meta-analysis of the efficiency and safety of alendronate versus teriparatide for treatment glucocorticoid-induced osteoporosis: A meta-analysis and systematic review of randomized controlled trials. They presented that teriparatide is an effective drug to reduce the risk of vertebral fracture in patients with GIOP. Furthermore, long-term use of teriparatide can increase the bone mineral density of lumbar spine, femoral neck, and total hip. This review is interesting. However, there are some concerns that need to be addressed.

1. Grey literature: If the authors performed the search using specific gray literature, please provide the name of such database. Overall, I have concerns about the reproducibility of the meta-analysis. There are some key flaws that may affect the summary effect estimate. Since search limited, unpublished studies were not included, I worry that this summary effect estimate will shrink or be nullified if grey literature, unpublished studies are found. The authors may only be able to say that their meta-analysis resolves discrepancies between published studies. I am hesitant because there are too many examples of unpublished data nullifying significant effects in meta-analyses. If the authors conduct a thorough search for unpublished data, this comment becomes irrelevant. Please add this information in supplementary.

2. The protocol registration was absent. Prospective registration of systematic reviews promotes transparency, helps reduce potential for bias and serves to avoid unintended duplication of reviews.

3. In "Methods" part, the independent investigators who extracted data and evaluated quality of each study should be clarify.

4. I wonder if the search strategies have been developed with the help of a librarian or experienced reviewers in the field. Generally, if the search yield is too low, systematic reviewers would need to modify the search term to ensure that it well cover most of the related papers. This could be done by in the PICO such as Control and Outcome; employ a free text rather than thesaurus search terms. I think the keywords for the outcome domain were not comprehensive enough to capture all potential synonyms of the outcomes of interested in this study, and therefore would be best not to limit the search with these terms. An example of the search strategies used for a particular database would show a transparency in this step (I don’t think the keywords presented is sufficient as an example of a search strategies as recommended in PRISMA).

5. The date of the literature was March 2021. In accordance with guidelines, the literature search should be performed until six months before the submission of the manuscript for publication. If the manuscript is accepted for publication, it is already outdated. In addition, the author should describe exact date and duration of the literature search in both abstract and main manuscript.

6. How authors dealt with missing data. Did you receive all answers from authors of the studies or make some imputation? In general, if the study did not report the data of the primary or secondary outcomes measures, the authors should contact via email to provide this information. Have you considered to contact authors via researchers’ network ResearchGate (https://www.researchgate.net/), Academia (https://www.academia.edu/), Loop (https://loop.frontiersin.org/) or Quora (https://es.quora.com/)? Nowadays these platforms are very useful and efficient canals to contact authors.

7. The authors should demonstrate both statistics and visualization. In addition, I suggest plot the funnel and contour-enhanced funnel in the graphic and not only the studies for better interpretation. Besides, it is necessary to present the p value for this analysis.

9. Finally, since I am not a native English user, I did not check for typo and grammatical errors thoroughly. This should be done by an appropriate language reviewer.

Reviewer #2: My comments and suggestions.

1. The statement in Introduction "Taking 2.5mg of oral prednisone increases the risk of fractures and when the dose is greater than 7.5mg, the risk increases by 5 times". Do you mean to say "the risk of fractures increased by 5 times with a dose of 7.5mg compared to 2.5mg"? This statement has to be fixed.

2. In the same statement - "the approximate physical number of endogenous glucocorticoids" is not clear. I would suggest changing to something like "dosage equivalent to daily endogenous glucocorticoid production".

3. Repetitive statements/conclusion: "The dose and duration of glucocorticoids have a significant effect on the occurrence of fractures. The daily average dose of glucocorticoids seems to be more predictive of the risk of fracture risk than cumulative doses." gets repeated 2 lines later as "In addition, the risk of fractures increases significantly with the increasing dose of glucocorticoids and the prolongation of time". Please avoid repetitions of the same fact.

4. Repetitive statements: "Long-term treatment and high-dose of glucocorticoids will cause osteoporotic fractures in 30-50% of adult patients". This statement is repetitive and was used 4 lines above under "In patients who receive long-term glucocorticoid treatment, 30-50% may have fractures, especially the lumbar spine (LS) femoral neck (FN) and total hip (TH)". Please avoid repetitions of the same fact.

5. I believe the mechanism of GIOP and pathogenesis is discussed in great detail but it does not relate to this paper at all. I would recommend shortening the pathogenesis/mechanism part in introduction. Otherwise consider adding a statement as to how the pathogenesis/mechanism is relevant to your paper.

6. In Introduction, page 6, the authors talk about bisphosphonates. "it has been proved to be an effective method for prevention and treatment of GIOP". Recommend removing "method". I would suggest changing it to "It has been shown to be effective for prevention and treatment of GIOP"

7. Same paragraph, page 6. "ALE is a kind of bisphosphonate". Remove kind of. It should be "ALE is a bisphosphonate"

8. What is the purpose of this statement "Decreased osteogenesis is the main pathogenesis of GIOP and glucocorticoids can significantly inhibit osteoblasts activity"?. You have a already discussed the mechanism/pathogenesis in the previous paragraph. its repetitive.

9. I recommend that the the authors discuss in the introduction on what's lacking in the current literature which prompted them to do this meta-analysis.

10. I see at least 2 meta-analysis on the same topic with very similar results. The authors mention these studies in their discussion. However they fail to add how is their study different from these other meta-analyses and how does their meta-analyses contribute to the current literature. I would recommend expanding more as discussion is very short.

10. Under discussion, authors talk about mechanism of ALE. Again, they discuss the mechanism of action of bisphosphonates under introduction. I would recommend removing this in the discussion or add a statement as to how is it relevant to your study. Because, the authors randomly state the mechanism of ALE during the discussion and then revert back to talking about prior meta-analyses results. There is no continuity.

11. The discussion should be well structured - I would recommend talking summarizing their results in the first paragraph. Then talk about prior studies that have compared these 2 drugs and summarize their results. Then talk about how this study is different/unique when compared to previous studies and what does it add to the literature. Then mention your strengths/limitations. I do not see a reason as to why mechanism of ALE and TPTD have been mentioned again in discussion when they already been mentioned in Introduction. Discussion NEEDS to be expanded.

12. Under study inclusion and exclusion criteria: "Patients were required to have a LS or TH

BMD T score of ≤−2.0 or ≤−1.0 plus at least one fragility fracture while taking glucocorticoids". According to standard definitions, Osteoporosis is defined as patients with BMD T score of ≤−2.5. Osteopenia is T score between -2.5 to -1.0. Why was -2.0 chosen here as the inclusion criteria and not -2.5?

13. Under results "MEAN PERCENT CHANGE FROM BASELINE IN THE BMD AT THE LS, FN, TH)". Authors have repeated the same results twice. "We further analyzed that the longer the time of taking TPTD within a certain period of time, the more obvious the 15 increase of LS BMD, and the LS BMD at 18 months was significantly higher than that at 6 months and 12 months. And the effect of TPTD on bone tissue was also different. For example, after taking TPTD for 18 months, the increase of BMD of LS was higher than that of FN and TH." The next line is repetition of this statement. "We found that the longer the time of taking TPTD within a certain period of time, the more obvious the increase of LS BMD. The LS BMD at 18 months was significantly higher than that at 6 months and 12 months. In addition, TPTD had different effects on different parts’ bone tissue. After taking TPTD for 18 months, the increase of bone mineral density of LS was higher than that of FN and TH."

6. PLOS authors have the option to publish the peer review history of their article (what does this mean?). If published, this will include your full peer review and any attached files.

Reviewer #1: **Yes: **Wisit Kaewput

Reviewer #2: No

---

## [Author Response · Author response to Decision Letter 0]

14 Mar 2022

Relevant files have been uploaded

---

## [Decision Letter · Decision Letter 1]

1 Apr 2022

PONE-D-22-00100R1The efficiency and safety of alendronate versus teriparatide for treatment glucocorticoid-induced osteoporosis: A meta-analysis and systematic review of randomized controlled trialsPLOS ONE

Dear Dr. Fei Yin,

Thank you for submitting your manuscript to PLOS ONE. After careful consideration, we feel that it has merit but does not fully meet PLOS ONE’s publication criteria as it currently stands. Therefore, we invite you to submit a revised version of the manuscript that addresses the points raised during the review process.

ACADEMIC EDITOR: Our expert reviewer(s) have recommended some minor revisions to your manuscript. Please also include timeline of the literature search in the method section of the abstract.

Therefore, I invite you to respond to the reviewer(s)' comments as below and revise your manuscript.

We look forward to receiving your revised manuscript.

Kind regards,

Wisit Cheungpasitporn, MD

Academic Editor

PLOS ONE

Journal Requirements:

Reviewers' comments:

Reviewer's Responses to Questions

**Comments to the Author**

1. If the authors have adequately addressed your comments raised in a previous round of review and you feel that this manuscript is now acceptable for publication, you may indicate that here to bypass the “Comments to the Author” section, enter your conflict of interest statement in the “Confidential to Editor” section, and submit your "Accept" recommendation.

Reviewer #1: (No Response)

Reviewer #2: All comments have been addressed

2. Is the manuscript technically sound, and do the data support the conclusions?

Reviewer #1: Yes

Reviewer #2: Yes

3. Has the statistical analysis been performed appropriately and rigorously? 

Reviewer #1: Yes

Reviewer #2: I Don't Know

4. Have the authors made all data underlying the findings in their manuscript fully available?

Reviewer #1: Yes

Reviewer #2: Yes

5. Is the manuscript presented in an intelligible fashion and written in standard English?

Reviewer #1: Yes

Reviewer #2: Yes

6. Review Comments to the Author

Reviewer #1: Did the search strategy the same in all databases? Search terms in PubMed and Embase are different. The authors should attach syntax used in each database in supplementary.

Reviewer #2: All of my comments have been addressed by the authors. I do not have any further recommendations or suggestions.

7. PLOS authors have the option to publish the peer review history of their article (what does this mean?). If published, this will include your full peer review and any attached files.

Reviewer #1: **Yes: **Wisit Kaewput

Reviewer #2: No

---

## [Decision Letter · Decision Letter 2]

14 Apr 2022

The efficiency and safety of alendronate versus teriparatide for treatment glucocorticoid-induced osteoporosis: A meta-analysis and systematic review of randomized controlled trials

PONE-D-22-00100R2

Dear Dr. Fei Yin,

We’re pleased to inform you that your manuscript has been judged scientifically suitable for publication and will be formally accepted for publication once it meets all outstanding technical requirements.

Kind regards,

Wisit Cheungpasitporn, MD

Academic Editor

PLOS ONE

Additional Editor Comments (optional):

It appears that all comments have been appropriately responded to. I recommend publication.

Reviewers' comments:

Reviewer's Responses to Questions

**Comments to the Author**

1. If the authors have adequately addressed your comments raised in a previous round of review and you feel that this manuscript is now acceptable for publication, you may indicate that here to bypass the “Comments to the Author” section, enter your conflict of interest statement in the “Confidential to Editor” section, and submit your "Accept" recommendation.

Reviewer #1: All comments have been addressed

Reviewer #2: All comments have been addressed

2. Is the manuscript technically sound, and do the data support the conclusions?

Reviewer #1: Yes

Reviewer #2: Yes

3. Has the statistical analysis been performed appropriately and rigorously? 

Reviewer #1: Yes

Reviewer #2: I Don't Know

4. Have the authors made all data underlying the findings in their manuscript fully available?

Reviewer #1: Yes

Reviewer #2: Yes

5. Is the manuscript presented in an intelligible fashion and written in standard English?

Reviewer #1: Yes

Reviewer #2: Yes

6. Review Comments to the Author

Reviewer #1: The authors addressed all my previous concerns and significantly improved quality of the manuscript. I have no additional comment.

Reviewer #2: No further comments. All the comments have been addressed by the authors extensively. I thank them for editing the article.

7. PLOS authors have the option to publish the peer review history of their article (what does this mean?). If published, this will include your full peer review and any attached files.

Reviewer #1: **Yes: **Wisit Kaewput

Reviewer #2: No

---

## [Editor Report · Acceptance letter]

19 Apr 2022

PONE-D-22-00100R2 

The efficiency and safety of alendronate versus teriparatide for treatment glucocorticoid-induced osteoporosis: A meta-analysis and systematic review of randomized controlled trials 

Dear Dr. Yin:

I'm pleased to inform you that your manuscript has been deemed suitable for publication in PLOS ONE. Congratulations! Your manuscript is now with our production department. 

Kind regards, 

on behalf of

Dr. Wisit Cheungpasitporn 

Academic Editor

PLOS ONE